# TET Family Members Are Integral to Porcine Oocyte Maturation and Parthenogenetic Pre-Implantation Embryogenesis

**DOI:** 10.3390/ijms241512455

**Published:** 2023-08-05

**Authors:** Fan Chen, Ming-Guo Li, Zai-Dong Hua, Hong-Yan Ren, Hao Gu, An-Feng Luo, Chang-Fan Zhou, Zhe Zhu, Tao Huang, Yan-Zhen Bi

**Affiliations:** 1Hubei Key Laboratory of Animal Embryo Engineering and Molecular Breeding, Institute of Animal Husbandry and Veterinary, Hubei Academy of Agricultural Sciences, Wuhan 430070, China; fanchen@hbaas.com (F.C.); 16699230598@163.com (M.-G.L.); zaidonghua@163.com (Z.-D.H.); 18971609780@163.com (H.-Y.R.); guhao@hbaas.com (H.G.); laf47630365@163.com (A.-F.L.); zhouchangfan@webmail.hzau.edu.cn (C.-F.Z.); zhuzhe_871230@163.com (Z.Z.); 2College of Animal Science and Technology, Shihezi University, Shihezi 832061, China

**Keywords:** pig, oocyte, embryo, 5-hydroxymethylcytosine, 5-methylcytosine, TET proteins, Bobcat339

## Abstract

The ten-eleven translocation (TET) enzyme family, which includes TET1/2/3, participates in active DNA demethylation in the eukaryotic genome; moreover, TET1/2/3 are functionally redundant in mice embryos. However, the combined effect of TET1/2/3 triple-gene knockdown or knockout on the porcine oocytes or embryos is still unclear. In this study, using Bobcat339, a specific small-molecule inhibitor of the TET family, we explored the effects of TET enzymes on oocyte maturation and early embryogenesis in pigs. Our results revealed that Bobcat339 treatment blocked porcine oocyte maturation and triggered early apoptosis. Furthermore, in the Bobcat339-treated oocytes, spindle architecture and chromosome alignment were disrupted, probably due to the huge loss of 5-hydroxymethylcytosine (5hmC)and concurrent increase in 5-methylcytosine (5mC). After Bobcat339 treatment, early parthenogenetic embryos exhibited abnormal 5mC and 5hmC levels, which resulted in compromised cleavage and blastocyst rate. The mRNA levels of EIF1A and DPPA2 (ZGA marker genes) were significantly decreased, which may explain why the embryos were arrested at the 4-cell stage after Bobcat339 treatment. In addition, the mRNA levels of pluripotency-related genes OCT4 and NANOG were declined after Bobcat339 treatment. RNA sequencing analysis revealed differentially expressed genes in Bobcat339-treated embryos at the 4-cell stage, which were significantly enriched in cell proliferation, cell component related to mitochondrion, and cell adhesion molecule binding. Our results indicated that TET proteins are essential for porcine oocyte maturation and early embryogenesis, and they act by mediating 5mC/5hmC levels and gene transcription.

## 1. Introduction

DNA methylation of cytosine at the C5 position (5-methylcytosine, 5mC) is an inhibitory epigenetic mechanism, which can block the expression of various genes [1]. The formation of 5mC is a dynamic process; ten-eleven translocation (TET) methylcytosine dioxygenase can successively oxidize 5mC to 5-hydroxymethylcytosine (5hmC), 5-formylcytosine (5fC), and 5-carboxylcytosine (5caC) [2]. The 5fC and 5caC can be replaced with unmodified cytosine via base excision repair [3]. Therefore, TET can eliminate DNA methylation imprinting and promote active demethylation processes. TET-mediated DNA demethylation is implicated in diverse biological processes, including embryotic development and stem cell pluripotency and differentiation [4]. Importantly, 5hmc serves not only as the first and important intermediate in demethylation processes but also an independent and stable epigenetic marker vital for stem cell pluripotency [5].

Global DNA demethylation occurs during two different stages of mammalian development, the migration of primordial germ cells (PGCs) and pronuclear stage of zygotes [6,7]. It seems that TET1, TET2, and TET3 act at different stages during embryo development. TET1 and TET2 predominantly participate in DNA demethylation in PGCs and post-implantation embryos [8,9]. TET3 mediates demethylation of both paternal and maternal genomes in zygotes and directly participates in active DNA demethylation in preimplantation embryos [10,11,12]. Moreover, 5hmC is present in fully grown oocytes but not at the nongrowing stage, which is accompanied by a noticeable TET3 increase; this implies that 5mC/5hmC conversion in oocytes is the initial step of embryotic demethylation [13]. Unexpectedly, even TET3 plays important roles in the methylation reprogramming of early embryos; TET3 is dispensable for oocyte maturation and early embryonic development, as viable mice pups could still be recovered after genetic ablation of TET3 [14]. TET1 or TET2 null mice or those with combined loss of two developed relatively normally [15], whereas TET1/2/3 triple-knockout was lethal for mouse embryos [16]. Therefore, TET1, TET2, and TET3 are functionally redundant in embryo development [17].

DNA methylation inhibits gene transcription; therefore, it is believed that global demethylation in preimplantation embryo contributes to general transcriptional change. Indeed, after fertilization, the zygote is transcriptionally dormant, and molecules essential for development (such as proteins and RNAs) are provided from the maternal oocyte. Subsequently, throughout epigenetic reprogramming, the maternal factors are degraded, and zygotic genome is activated. This developmental switch is termed as zygotic gene activation (ZGA). In mice, ZGA is initiated during the late 1-cell stage, followed by major gene activation (major ZGA) at the 2-cell stage [18]. In pigs, major ZGA occurs at the 4-cell stage [19]. In humans, major ZGA occurs at the 8-cell stage [20]. In embryos of various species, TET-mediated DNA demethylation is crucial to initiate and complete ZGA [21,22]. Moreover, compared with fertilized embryos, cloned embryos exhibit abnormal 5hmC level and insufficient ZGA initiation, which are partially caused by the deficiency of TET protein [23,24].

As TET proteins have dynamic and overlapping roles in embryos, it is a challenge to characterize the function of individual TET protein. The effect of TET1/2/3 knockdown or knockout is not studied in porcine embryo [25,26]. Bobcat339, a promising and novel cytosine-based selective TET enzyme inhibitor, can reduce 5hmC levels in DNA [27]. Therefore, in the present study, Bobcat339 was used to explore how the combined suppression of these three members affects oocyte maturation and embryo development in pig. Our results indicated that Bobcat339, as a special TET family inhibitor, compromised porcine oocyte maturation and early embryotic development.

## 2. Results

### 2.1. Bobcat339 Treatment Obstructed the First Polar Body Extrusion in Porcine Oocytes

To investigate the influence of TET on porcine cumulus cell expansion and oocyte maturation, cumulus oocyte complexes (COCs) were cultured with increasing concentrations of Bobcat339 (0, 100, 200, and 400 μM), and the responses of COCs were recorded at 24, 36, and 44 h. Bobcat339 treatment could block the expansion of cumulus cells in time- and concentration-dependent manners (Figure 1A). Furthermore, the first polar body extrusion (PBE) was remarkably impaired in Bobcat339-treated oocytes in a concentration-dependent manner (71.75 ± 3.56%, n = 163, control vs. 57.00 ± 4.06%, n = 171, 100 μM, *p* < 0.01 vs. 32.50 ± 2.18%, n = 185, 200 μM, *p* < 0.001; vs. 13.00 ± 2.12%, n = 201, 400 μM, *p* < 0.001; Figure 1B). These results implied that oocyte maturation was hampered after the inhibition of TET by certain concentrations of Bobcat339. Since 200 μM Bobcat339 could significantly hamper the PBE but allowed escaping of some oocytes, this concentration was used for subsequent studies on the oocytes (Figure 1C).

### 2.2. Bobcat339 Treatment Triggered Apoptosis of Porcine Oocytes

With the weakened expansion of cumulus cells and PBE rate in Bobcat339-treated oocytes, we speculated that the apoptotic level of oocytes may be elevated after Bobcat339 treatment. Annexin-V-FITC staining and quantitative analysis revealed that the amount of early apoptotic oocytes was significantly increased after Bobcat339 treatment (4.35 ± 0.88%, n = 76, control vs. 16.70 ± 3.83%, n = 91, 200 μM, *p* < 0.05; Figure 2A,B). Moreover, the mRNA levels of BAX and BCL-2 were significantly higher and the BAX/BCL-2 ratios were remarkably elevated in the Bobcat339-treated oocytes (Figure 2C). Therefore, we concluded that inhibition of TET proteins triggered early apoptosis in porcine oocytes.

### 2.3. Effect of Bobcat339 on Spindle Architecture and Chromosomes Alignment in Oocytes

Bobcat339-treated oocytes exhibited defects in maturation and early apoptosis. Therefore, we speculated that the spindle architecture could be disrupted after Bobcat339 treatment. Therefore, the spindle morphology was assessed at metaphase I (MI) stage in control and Bobcat339-treated oocytes using α-tubulin staining. Our results revealed that the control oocytes had a typical barrel spindle, whereas the spindles were disorganized in the Bobcat339-treated oocytes. Moreover, unlike the control oocytes, in which the chromosomes were well aligned at the spindle plate, Bobcat339-treated oocytes exhibited dispersed chromosomes (Figure 3A). The percentage of aberrant spindles in Bobcat339-treated oocytes was significantly elevated compared with that in the control oocytes (30.01 ± 4.97% vs. 14.33 ± 3.40%, n = 123, *p* < 0.05; Figure 3B). In addition, the proportion of misaligned chromosomes was increased in Bobcat339-treated oocytes (66.32 ± 7.36% vs. 17.67 ± 1.70%, n = 145, *p* < 0.001; Figure 3C). These results indicated that TET family is implicated in spindle architecture and chromosome alignment in porcine oocytes.

### 2.4. Bobcat339 Treatment Altered 5mC/5hmC Levels in Porcine Oocytes

To assess whether the above defects depend on the enzymatic activity of TET proteins, we examined the 5mC and 5hmC levels in the control and Bobcat339-treated oocytes. The results demonstrated that the fluorescent signal of 5hmC was significantly decreased in the Bobcat339-treated oocytes (Figure 4A). Further quantitative fluorescence analysis confirmed that 5hmC signal intensity was significantly decreased in Bobcat339-treated oocytes (Figure 4B). On the contrary, the 5mC signal was significantly increased in the Bobcat339-treated oocytes (Figure 4C). Quantitative analysis further verified that the mean 5mC intensity was significantly elevated in Bobcat339-treated oocytes (Figure 4D). These results suggested that TET family mediates the levels of both 5mC and 5hmC in porcine oocytes.

### 2.5. Effect of Bobcat339 on Preimplantation Embryo Development

To assess the impact of TET on early embryonic development, porcine parthenogenetic embryos were cultured in a PZM-3 medium supplemented with increasing concentrations of Bobcat339 (0, 25, 50, and 100 µM). We recorded the blastocyst rates on day 6 (Figure 5A,B). The proportion of activated oocytes that developed to the blastocyst stage was significantly lower after Bobcat339 treatment (33.67 ± 3.01%, n = 178, control vs. 31.33 ± 0.02%, n = 164, 25 μM, *p* > 0.05 vs. 11.02 ± 3.07%, n = 215, 50 μM, *p* < 0.01 vs. 2.34 ± 2.00%, n = 237, 100 μM, *p* < 0.001). In addition, we calculated the developmental rate of embryos on day 3 after activation and observed that no significant difference was observed between the two groups at the 2-cell stage (7.76 ± 3.16%, n = 204, control vs. 10.82 ± 2.48%, n = 234, 25 μM, *p* > 0.05 vs. 13.73 ± 4.00%, n = 219, 50 μM, *p* > 0.05 vs. 11.75 ± 2.37%, n = 194, 100 μM, *p* > 0.05; Figure 5C); however, the developmental rate of embryos was significantly increased at the 4-cell stage (9.59 ± 1.50% control vs. 19.85 ± 3.22%, 25 μM, *p* < 0.05 vs. 29.36 ± 3.77%, 50 μM, *p* < 0.001 vs. 42.82 ± 5.06%, 100 μM, *p* < 0.01; Figure 5C) and decreased at 8-cell stage (76.68 ± 3.67% control vs. 63.88 ± 3.60%, 25 μM, *p* < 0.05 vs. 53.93 ± 4.87%, 50 μM, *p* < 0.05 vs. 37.90 ± 7.05%, 100 μM, *p* < 0.001) after Bobcat339 treatment. Therefore, Bobcat339 treatment arrested embryo growth at the 4-cell stage.

### 2.6. Bobcat339 Treatment Decreased the Expression of ZGA and Pluripotency-Related Genes

Bobcat treatment arrested embryo growth at the 4-cell stage. However, porcine ZGA occurs during the 4-cell stage. Therefore, the effect of Bobcat339 treatment on the expression of porcine ZGA marker genes (EIF1A, DPPA2, and ZSCAN4) was evaluated using qRT-PCR. The results indicated that the mRNA levels of EIF1A and DPPA2 at the 4-cell stage were significantly lower after Bobcat339 treatment (Figure 6A). As pluripotency-related and imprinted genes are crucial for embryotic development, we assessed the mRNA levels of the pluripotency-related genes SOX2, OCT4, and NANOG and imprinted gene H19. The mRNA levels of OCT4 and NANOG significantly decreased after Bobcat339 treatment (Figure 6B), whereas those of SOX2 and H19 exhibited no significant difference (Figure 6C). These findings verified that Bobcat339 treatment could disrupt ZGA in porcine embryos.

### 2.7. Bobcat339 Treatment Disrupted 5mC/5hmC Levels in Porcine Embryos

To investigate whether the embryonic developmental arrest after Bobcat339 treatment occurred via the enzymatic activity of TET, we examined the 5mC/5hmC levels in control and Bobcat339-treated preimplantation embryos. The fluorescence intensity of 5hmC in 2- and 4-cell embryos was lower in Bobcat339-treated groups (Figure 7A,B). Further quantitative fluorescence analysis confirmed that the fluorescence intensity of 5hmC was significantly decreased after Bobcat339 treatment (Figure 7E,F). On the contrary, the fluorescence signal of 5mC in 2- and 4-cell embryos was higher in Bobcat339-treated groups (Figure 7C,D). The fluorescence intensity of 5mC was significantly elevated in the Bobcat339-treated groups (Figure 5G,H). These results suggested that TET enzymes regulated both 5mC and 5hmC levels in porcine preimplantation embryos.

### 2.8. Comparative Analyses of Transcriptomic Data

To delineate the affected genes and pathways after Bobcat339 treatment, RNA sequencing analysis was performed in 4-cell embryos derived from control and Bobcat339-treated groups. In our results, principal component analysis (PCA) met the requirement of mRNA expression analysis (Figure 8A). Violin plot was not significantly different between control and Bobcat339-treated embryos (Figure 8B). Hierarchical clustering of the differentially expressed genes (DEGs) in control and Bobcat339-treated groups is shown in Figure 8C. Furthermore, we identified 203 significantly DEGs (75 up- and 128 downregulated genes) between the two groups (Figure 8D). The GO analysis revealed that the DEGs were significantly enriched in biological processes including regulation of cell population proliferation (GO:0042127), in molecular functions related to cell adhesion molecule binding (GO:0050839), and cellular components related to mitochondrion (GO:0005739) and organelle envelope (GO:0031967) (Figure 8E). Therefore, Bobcat339 treatment significantly altered the transcriptome in the embryos at the 4-cell stage.

## 3. Discussion

It is reported that TET1 and TET2 are mainly responsible for DNA demethylation in PGCs and post-implantation embryos, and TET3 is implicated in active demethylation of DNA in the zygote. However, several studies have reported that TET1, TET2, and TET3 probably have overlapping expressions and roles in mice embryo development [28]. TET1/TET2 double deficiency embryos could survive normally and develop into fertile adult mice, likely because TET3 compensates for TET1/TET2 mutation; however, TET1/2/3 triple deficient embryos were arrested at the 2-cell stage [15,29]. These data highlighted that a compensatory mechanism exists among TET1, TET2, and TET3. Therefore, in this study, we used Bobcat339, a novel and specific small-molecular inhibitor of the TET proteins, to explore the combined role of TET proteins in porcine oocytes and preimplantation embryos.

In this study, the levels of 5mC and 5hmC were abnormal after Bobcat339 treatment, with elevated 5mC and declined 5hmC levels in porcine oocytes. This phenotype is different from that of mice oocytes with only TET3 deficiency, in which the reduction in the 5hmC levels or increase in the 5mC levels was not apparent compared with the control oocytes [14]. Assessment of apoptotic signal and apoptosis-related genes revealed early apoptosis after Bobcat339 treatment, in accordance with the results in glioma cells [30]; correspondingly, the PBE rates decreased. Moreover, Bobcat339-treated oocytes exhibited remarkably abnormal spindle architecture and chromosome alignment. These phenotypes resembled those of mice oocytes with TET2 deficiency, in which the defects of spindle morphology and chromosome distribution, delayed meiotic progression, and accelerated aging were observed [31].

In this study, we observed that after Bobcat339 treatment, the levels of 5mC and 5hmC in parthenogenetic embryo were altered. These results are consistent with previous studies on mice, bovine, and human embryos with TET1/2/3 triple-knockout [32,33]. 5hmC is not only an intermediate between 5mC and cytosine but also an independent epigenetic marker vital for the pluripotency [5]. Pluripotency-associated genes such as OCT4, SOX2, and NANOG are methylated during germ cell development, and their demethylation occurs in the early embryo. In normal zygotes, pluripotency-associated genes undergo substantial demethylation till the PN3–4 pronuclear stages [34]. The impacts of individual TET family member on the pluripotency of embryos are not consistent in previous studies [35,36,37]. This could be due to the compensatory role played by TET family members. Our results indicated that the inhibition of TET resulted in noticeably downregulated expression of OCT4 and NANOG.

Imprinted region is referred to the alleles expressed only from paternal or maternal origin. They are distinguished by the differences in DNA methylation. Unlike pluripotency-associated genes, imprinted genes are resistant to demethylation in the early embryo stage and are protected by maternal factors including Dnmt1, Dppa3, ZFP57, and TRIM28 [38]. It is well known that parthenogenetic embryos naturally exhibit abnormal expression of imprinted genes, in which paternal imprinted genes are highly expressed, and maternal imprinted genes are either lowly or not expressed [39]. Therefore, parthenogenetic embryo is the most appropriate model for exploring the methylation modifications. In parthenogenetic embryos, H19 is an important paternal imprinted gene whose expression level depends on methylation regulation [40]. In our study, the mRNA level of H19 was not affected in the Bobcat339-treated embryos. This is consistent with a study on bovine parthenogenetic embryo [32].

In pigs, the major ZGA occurs during the 4-cell stage [19]. In our study, the growth of Bobcat339-treated embryos was arrested at the 4-cell stage. Moreover, transcriptome data of early embryos at the 4-cell stage revealed 203 significant DEGs between Bobcat339-treated and control groups, with 75 up- and 128 downregulated genes (The DEGs in Bobcat339 groups compared to Control are shown in Appendix A). Therefore, after Bobcat339 treatment, abnormal 5mC/5hmC levels may lead to altered ZGA, finally blocking embryotic development. Therefore, we assessed the mRNA levels of porcine ZGA marker genes EIF1A, DPPA2, and ZSCAN4 [41]. The significantly decreased expression of EIF1A and DPPA2 confirmed our hypothesis.

## 4. Materials and Methods

### 4.1. Antibodies and Reagents

Rabbit anti-5hmC monoclonal antibody (Cat# 91309) was obtained from Active Motif Company (Carlsbad, CA, USA); rabbit anti-5mC monoclonal antibody (Cat# ab214727) was purchased from Abcam Company (Boston, MA, USA); mouse anti-α-tubulin-FITC antibody (Cat# F2168) was obtained from Sigma Chemical Company (St Louis, MO, USA); DyLight 549-conjugated goat anti-rabbit IgG (H + L) was purchased from Abbkine Biotechnology (San Diego, CA, USA); FITC-conjugated goat anti-rabbit IgG (H + L) was purchased from Boster (Wuhan, China); Bobcat339 (>99.9%) (Cat# S6682) was obtained from Selleck Chemical Company (Pittsburgh, PA, USA). All other chemicals and culture media were purchased from Sigma Chemical Company (St Louis, MO, USA) unless otherwise stated.

### 4.2. Porcine Oocyte Collection and In Vitro Maturation (IVM)

The porcine ovaries were obtained from a local slaughterhouse in Wuhan and transported to the laboratory while maintained at 38.5 °C in 0.9% saline. Follicular fluid was prepared from 3 to 6 mm antral follicles using a 10 mL syringe with a 16-gauge needle. Cumulus oocyte complexes (COCs) with uniform cytoplasm and several layers of cumulus cells were selected and rinsed three times in TCM-199 medium with 10% porcine follicular fluid (pFF), 5 μg/mL insulin-transferrin-selenium, 10 ng/mL EGF, 0.6 mM L-cysteine, 0.2 mM pyruvate, 25 μg/mL kanamycin. Approximately 40 COCs per well were cultured in 4-well plates containing in 500 µL TCM-199 medium with 10% pFF, 5 μg/mL insulin-transferrin-selenium, 10 ng/mL EGF, 0.6 mM L-cysteine, 0.2 mM pyruvate, 25 μg/mL kanamycin and 5 IU/mL of each eCG and hCG, covered with 300 µL mineral oil.

### 4.3. Parthenogenetic Activation and In Vitro Culture (IVC)

Cumulus cells were removed by exposure to 0.2% hyaluronidase for 3 min. Oocytes with the first polar body were subjected to Parthenogenetic Activation (PA) using two DC pulses of 1.2 kV/cm for 30 ms in fusion medium (0.3 M D-mannitol supplemented with 0.175 mM CaCl2·2H2O and 0.05 mM MgCl2·6H2O). The electro-activated oocytes were incubated in PZM-3 containing 7.5 µg/mL cytochalasin B for 3 h. Next, they were washed and cultured in a 4-well dish containing 500 µL PZM-3 with 0.4% bovine serum albumin (BSA) at 38.5 °C under 5% CO_2_. The developmental rates were evaluated at 80 h (Day 3) and 124 h (Day 6) of IVC after activation.

### 4.4. Chemical Treatment

Bobcat339 was dissolved in DMSO (400 mM) and then diluted with maturation medium to the final concentrations of 0 μM, 50 μM, 100 μM, 200 μM, and 400 μM, with the concentration of DMSO < 0.1% in the culture medium. In addition, to arrest the porcine oocytes at the meiotic germinal vesicle (GV) stages, culture medium was supplemented with 1 mM IBMX (Sigma, 15879, St Louis, MO, USA).

### 4.5. AnnexinV-FITC Staining

Early apoptosis of the porcine oocytes was detected using the AnnexinV-FITC Apoptosis Detection Kit (Beyotime, C1062S, Shanghai, China). Briefly, 20~30 oocytes were washed in PBS containing 0.01% PVA (*w*/*v*), and then incubated in 100 μL binding buffer containing 5 μL of AnnexinV-FITC for 15 min at room temperature. After sufficient washing, the oocytes were checked immediately with inverted fluorescence microscopy.

### 4.6. RNA Isolation and RT-PCR

Total RNA was obtained from 50 oocytes or 10 4-cell stage embryos using RNAqueous Microkit (Thermo Fisher Scientific, Waltham, MA, USA) and was treated with DNase I (Thermo Fisher Scientific, Massachusetts, MA) to prevent genomic DNA contamination. Reverse transcription was performed using the RETROscript kit (Thermo Fisher Scientific, Waltham, MA, USA). The real-time PCR was carried out in an S1000 (Bio-Rad Laboratories, Hercules, CA, USA). Each of 25 μL reaction system contained 12.5 μL of SYBR Green supermix (BioRad Laboratories, Hercules, CA, USA), 0.5 μL of each primer, and 11.5 µL of cDNA. The 2^−ΔΔCT^ method was used to calculate relative expression levels. The primers used for real-time PCR reactions were included in Table 1.

### 4.7. Immunofluorescent Staining

Oocytes or embryos were first fixed with 4% paraformaldehyde for 30 min and then permeabilized with 1% Triton X-100 at room temperature for at least 8 h. For 5mC and 5hmC immunostainings, fixed embryos were incubated in 4N HCl solution at RT for 15 min, followed by neutralization (10 min, 100 mM Tris–HCl, pH 8.0) and the second fixation. Then, after blocking with 2% BSA supplemented washing buffer (0.1% Tween 20 and 0.01% Triton X-100 in PBS) for 1 h, oocytes or embryos were stained with 5mC or 5hmC primary antibodies at 4 °C overnight. After three washes in washing buffer, oocytes or embryos were stained with Cy3 or FITC-conjugated goat anti-rabbit immunoglobulin G (IgG) (H + L) (1:100) for 1 h at room temperature. For the α-tubulin-FITC staining, permeabilized oocytes were directly incubated for 1 h at room temperature after sufficient washing. Finally, oocytes or embryos were mounted on glass slides and examined with inverted fluorescence microscopy (Nikon Eclipse TE2000-U, Tokyo, Japan). Each experiment was repeated at least three times, and no less than 30 oocytes or embryos were examined for each.

### 4.8. RNA Sequencing and Data Analysis

The total RNA of each sample (100 4-cell embryos) was extracted using TRIzol^®^ Reagent (Invitrogen, Carlsbad, CA, USA), and the smart-seq2 library was constructed using a previously published protocol [42]. Briefly, the full-length cDNA was synthesized from the RNA using template switch oligo and SuperScript II reverse transcriptase (Invitrogen, Carlsbad, CA, USA). The PCR protocol was as follows: incubation at 42 °C for 90 min, followed by 10 cycles of 50 °C for 2 min and 42 °C for 2 min. The reverse transcriptase was inactivated by heating at 70 °C for 15 min. The single-stranded cDNA was subsequently amplified using Kapa HiFi HotStart Readymix (Roche, Basel, Switzerland). The amplified cDNA was purified using AMpure XP beads (Beckman, Brea, CA, USA) and used for library construction. All libraries were assessed using the High DNA Sensitivity Bioanalyzer 2100 (Agilent, Santa Clara, CA, USA) and real-time quantitative PCR (qRT-PCR). All libraries were sequenced using 150-bp paired-end on the Illumina Novaseq platform (Santa Clara, CA, USA).

Raw RNA was filtered using Trimmomatic, and the sus scrofa reference genome (NCBI.Sscrofa10.2) was obtained. Index of the reference genome was built using Hisat2 v2.0.5, and paired-end clean reads were aligned to the reference genome. Htseq-count was used to count the read numbers mapped to each gene. Differential expression analysis of Bobcat339-treated and control groups was performed using the DESeq2 R package. DEGs were determined with the criteria of *p* value < 0.05 and FoldChange > 2. Cluster analyses were performed using the K-means clustering algorithm with R. Gene Ontology (GO) enrichment analysis was performed using the topGO R package 4.1.0.

### 4.9. Statistical Analysis

At least three independent replicates were used for each analysis. Data were presented as mean ± SEM and analyzed by paired-samples *t*-test using GraphPad Prism8 analysis software, and *p* < 0.05 was considered to be statistically significant.

## 5. Conclusions

In this study, we demonstrated that TET family is essential for porcine oocyte maturation and participates in oocyte spindle architecture and chromosome alignment by regulating 5mC and 5hmC levels. Our results also indicated that TET family is essential for the development of parthenogenetic preimplantation embryos, mediating the mRNA level of ZGA marker genes and pluripotency-related genes and the global transcriptome of embryos at the 4-cell stage through adjusting 5mC and 5hmC levels.

## Figures and Tables

**Figure 1 ijms-24-12455-f001:**
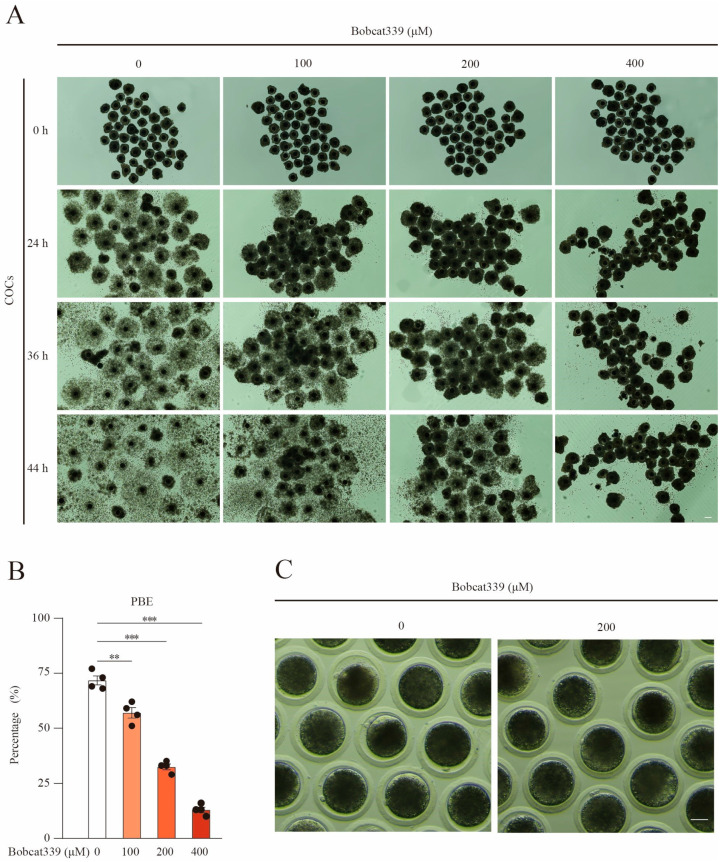
Bobcat339 treatment impaired porcine oocyte maturation. (**A**) Representative images of cumulus oocyte complexes (COCs) in control and Bobcat339-treated (0, 100, 200, and 400 μM) oocytes after culturing in vitro for 24, 36, and 44 h. Good expansion of cumulus cells was observed in control COCs, whereas they exhibited less expansion and were more adhesive in the Bobcat339-treated group. Scale Bar = 50 μm. (**B**) The percentage of the PBE was significantly decreased after treatment with 100 (** *p* < 0.01), 200 (*** *p* < 0.001), and 400 μM (*** *p* < 0.001) Bobcat339. The results are shown as the mean ± SEM from at least three independent experiments. (**C**) Representative images of PBE in control and 200 μM Bobcat339-treated oocytes after culturing in vitro for 44 h. Scale Bar = 50 μm.

**Figure 2 ijms-24-12455-f002:**
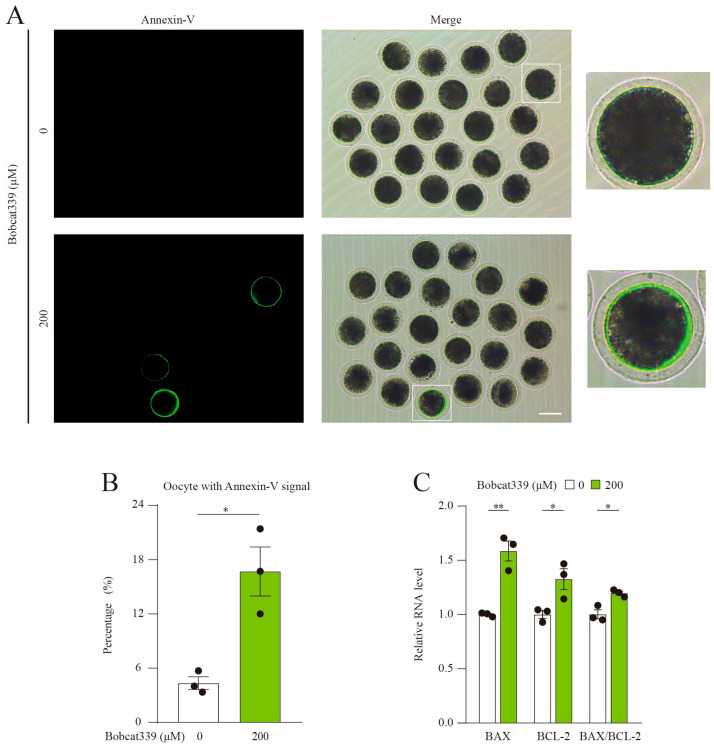
Bobcat339 treatment induced the early apoptosis of porcine oocytes. (**A**) Early apoptotic fluorescent signals in control and Bobcat339-treated oocytes. Annexin-V, green; Scale Bar = 100 μm. (**B**) The proportion of the oocytes with Annexin-V signal was calculated in control and Bobcat339-treated oocytes. (**C**) qRT-PCR revealed the mRNA levels of BAX, BCL-2 and BAX/BCL-2 in control and Bobcat339-treated oocytes. * *p* < 0.05; ** *p* < 0.01. The results are shown as the mean ± SEM from at least three independent experiments.

**Figure 3 ijms-24-12455-f003:**
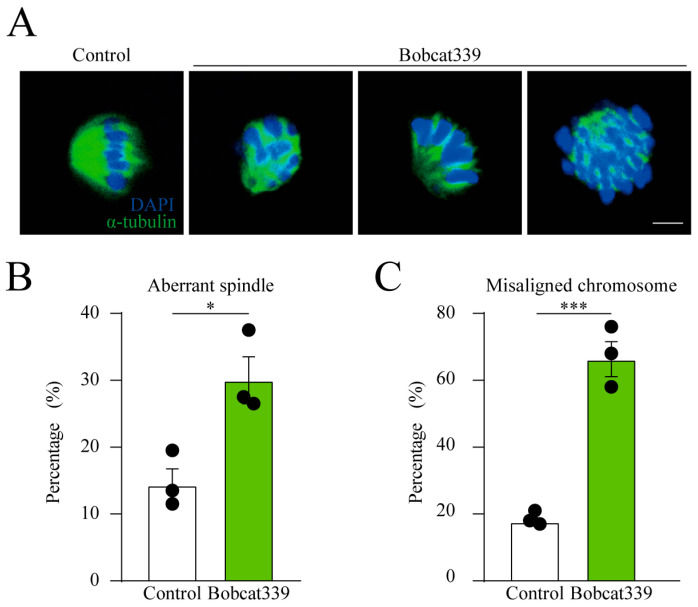
Influence of Bobcat339 on spindle morphology and chromosome alignment. (**A**) Images delineating spindle morphology and chromosome alignment in control and Bobcat339-treated oocytes (200 μM Bobcat339). Blue, chromosomes; Green, α-tubulin; Scale Bar = 20 μm. (**B**) The percentage of aberrant spindle was significantly increased in Bobcat339-treated oocytes. (**C**) The proportion of misaligned chromosome was significantly elevated in Bobcat339-treated oocytes. * *p* < 0.05; *** *p* < 0.001. The results are shown as the mean ± SEM from at least three independent experiments.

**Figure 4 ijms-24-12455-f004:**
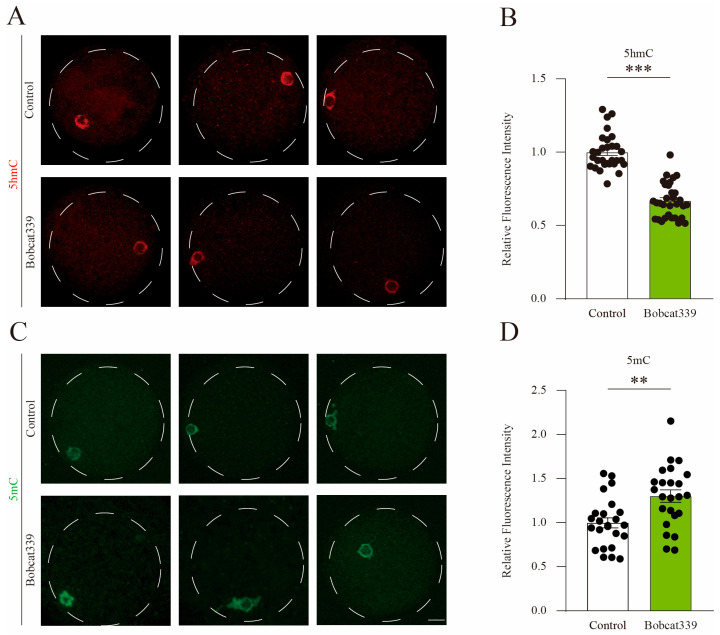
Bobcat339 treatment altered 5mC and 5hmC levels in porcine oocytes. The COCs in control and Bobcat339-treated groups were cultured in the IVM medium supplemented with 1 mM IBMX, to maintain the meiotic GV arrest. (**A**) The immunofluorescent staining of 5hmC in porcine oocytes. Red, 5hmC; Scale Bar = 20 μm. (**B**) The fluorescence intensity of 5hmC in control and Bobcat339-treated oocytes. (**C**) Immunofluorescent staining of 5mC in control and Bobcat339-treated oocytes. Green, 5mC; Scale Bar = 20 μm. (**D**) Average fluorescence intensity of 5mC in control and Bobcat339-treated oocytes. ** *p* < 0.01; *** *p* < 0.001. The results are shown as the mean ± SEM from at least three independent experiments.

**Figure 5 ijms-24-12455-f005:**
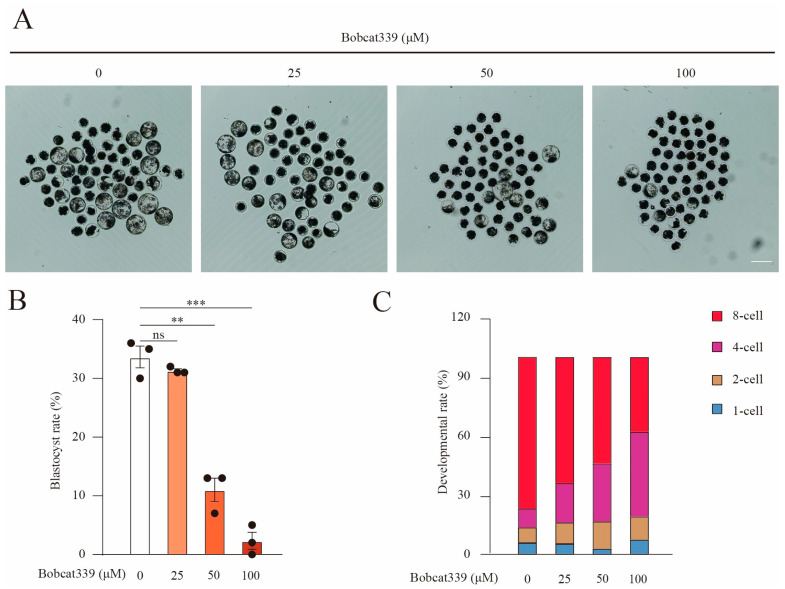
Bobcat339 treatment hampered early embryo development in pigs. (**A**) Images delineating blastocyst formation on day 6 of parthenogenetically activated porcine embryos in control and Bobcat339-treated (25, 50, and 100 μM) groups. Scale Bar = 100 μm. (**B**) The proportions of activated oocytes that developed to the blastocyst stage were calculated between control and Bobcat339-treated groups. (**C**) The proportions of embryos at various stages were calculated on day 3 in control and Bobcat339-treated (25, 50, and 100 μM Bobcat339) groups. ns (no significance) represents *p* > 0.05; ** *p* < 0.01; *** *p* < 0.001. The results are shown as the mean ± SEM from at least three independent experiments.

**Figure 6 ijms-24-12455-f006:**
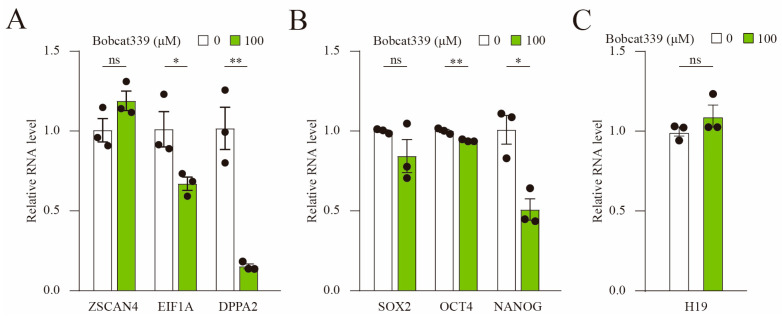
Bobcat339 treatment blocked the expression of ZGA and pluripotency genes. (**A**) The mRNA levels of EIF1A and DPPA2 were significantly decreased in Bobcat339-treated embryos. (**B**) The mRNA levels of OCT4 and NANOG were reduced in Bobcat339-treated embryos; the mRNA levels of SOX2 exhibited no change. (**C**) The mRNA levels of H19 were not affected after Bobcat339 treatment. ns (no significance) represents *p* > 0.05; * *p* < 0.05; ** *p* < 0.01. The results are shown as the mean ± SEM from at least three independent experiments.

**Figure 7 ijms-24-12455-f007:**
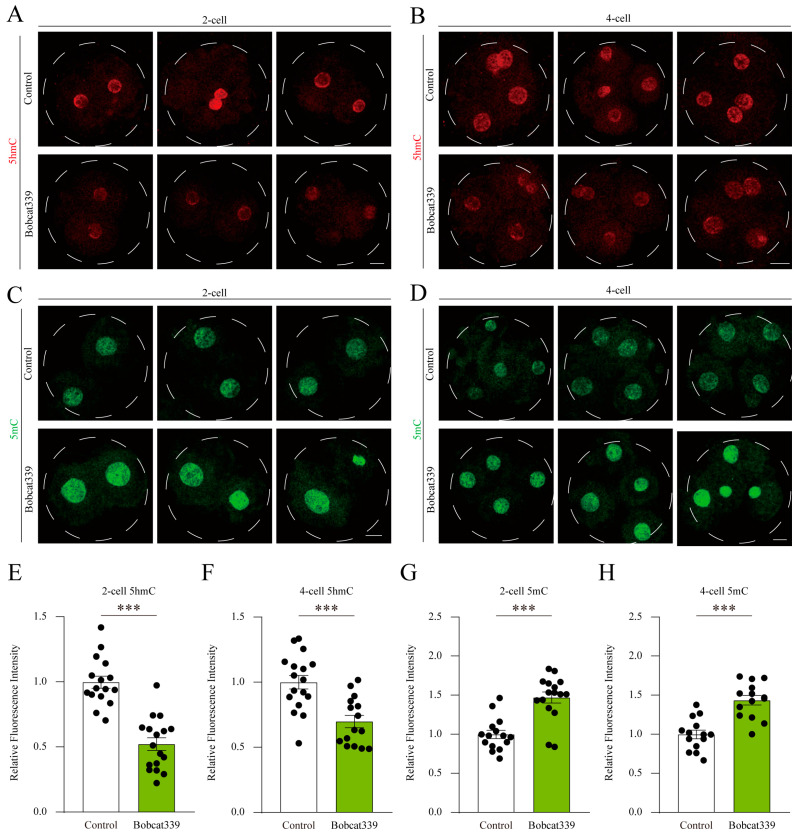
Bobcat339 treatment affected 5mC and 5hmC levels in porcine embryos. (**A**,**B**) The immunofluorescence staining of 5hmC in embryos at the 2- and 4-cell stages in control and Bobcat339-treated groups. Red, 5hmC; Scale Bar = 20 μm. (**C**,**D**) The immunofluorescence staining of 5mC in embryos at the 2- and 4-cell stages in control and Bobcat339-treated groups. Green, 5mC; Scale Bar = 20 μm. (**E**,**F**) The fluorescence intensity of 5hmC was calculated in embryos at the 2- and 4-cell stages in control and Bobcat339-treated groups. (**G**,**H**) Average fluorescence intensity of 5mC was recorded in embryos in control and Bobcat339-treated groups at the 2- and 4-cell stages. The results are shown as the mean ± SEM from at least three independent experiments. *** *p* < 0.001.

**Figure 8 ijms-24-12455-f008:**
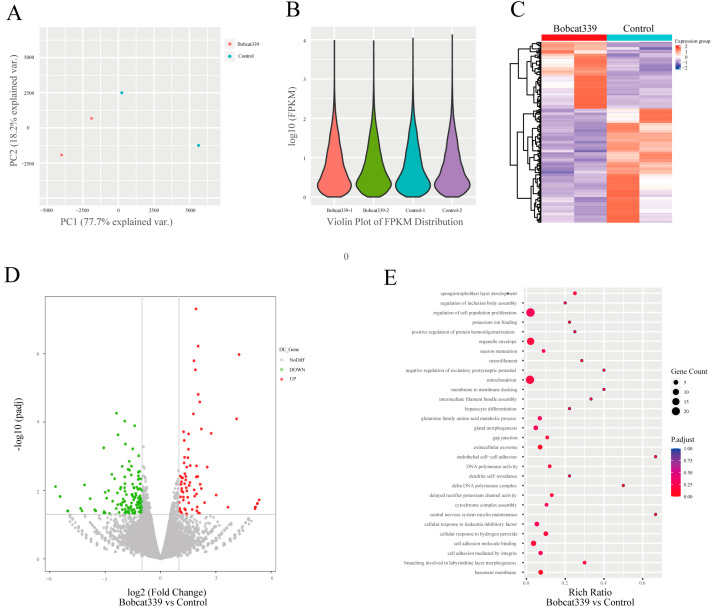
The different gene expression patterns between control and Bobcat339-treated embryos. (**A**) Principal component analysis of mRNA level between control and Bobcat339-treated embryos at the 4-cell stage. (**B**) Violin plot of gene expression and distribution between control and Bobcat339-treated groups. (**C**) Hierarchical cluster of differentially expressed genes (DEGs; *p* < 0.05 and |log2 fold change| > 1) between control and Bobcat339-treated embryos at the 4-cell stage. The color scale of the heatmap represents the expression levels. (**D**) Volcano plot of DEGs between control and Bobcat339-treated embryos at the 4-cell stage. Red and green dots represent up- and down regulated genes, respectively. (**E**) Gene Ontology analysis revealed that the DEGs were enriched in certain biological processes, molecular functions, and cellular components.

**Table 1 ijms-24-12455-t001:** Primer sequences used for real-time PCR.

Gene	Accession	Primer Sequences (5′ to 3′)	Product Size (bp)
BCL-2	XM_021077298.1	F: CAGGGACAGCGTATCAGAGC R: TTGCGATCCGACTCACCAAT	156
BAX	XM_013998624.2	F: CCAGGATCGAGCAGGGCGAAT R: CACAGGGCCTTGAGCACCAGTTT	285
DPPA2	XM_003358822.4	F: CCGTTCCTGCTTCTGTTGAGACC R: GGCGAACCCAACCTTCTGTATCTG	105
EIF1A	NM_001243218.1	F: GGTGTTCAAAGAAGATGGGCAAGAG R: TTTCCCTCTGATGTGACATAACCTC	115
H19	AY044827.1	F: TCAAACGACAAGAGATGGTGCTA R: GACGTCTGTTCCTTTGGCTC	118
NANOG	XM_021092390.1	F: AGGACAGCCCTGATTCTTCCACAA R: AAAGTTCTTGCATCTGCTGGAGGC	198
OCT4	XM_021097869.1	F: AAGCAGTGACTATTCGCAAC R: CAGGGTGGTGAAGTGAGG	136
RPLP0	NM_001129964.2	F: GCTAAGGTGCTCGGTTCTTC R: GTGCGGACCAATGCTAGG	112
SOX2	NM_001123197.1	F: CGCAGACCTACATGAACG R: TCGGACTTGACCACTGAG	103
TET1	NM_001315772.1	F: AGCACAGGACAAAATGAAGG R: TGGTTAGTTGGAGAGGAGG	171
TET2	XM_013978993.2	F: GCCAACCCTGTGAACCTCT R: GGGCTGGTAAAGTGTATGG	270
TET3	XM_021087365.1	F: TCAAGGCAAAGACCCGAAC R: AGACGGCAGTCAATCGCTATT	261
ZSCAN4	XM_021097584.1	F: GCCCAGAAAGTCTTCCCATGTGAG R: GCCTCTCATCATTGTGTCTCCTCTG	94

## Data Availability

The study was approved by the Institutional Review Board (or Ethics Committee) of the Hubei Academy of Agricultural Sciences (Approval ID: HBAAS (Hubei) 20230005).

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
