# Peer review of "TET Family Members Are Integral to Porcine Oocyte Maturation and Parthenogenetic Pre-Implantation Embryogenesis"

_ijms, 2023, doi:10.3390/ijms241512455_

Round 1

Reviewer 1 Report

The authors herein provide valuable experimental data on the effects of a novel TET family inhibitor on porcine maturation. Data reported suggested alteration of live/death balance ad and alteration in the embryo pluripotency as revealed by expression analyses.

A few points should be better discussed:

1. I prefer to avoid the use of "enzyme suppression". You are not confident about the percentage of activities remaining after using the inhibitor. So please change the term.

2. Bobcat 339 has been recently defined as probable apoptotic triggers at least in glioma cells. Please cite this recent abstract 10.1093/neuonc/noab196.654

3. Why do not confirm qPCR data (BAX and BCL2) via western blot? It is the first time that the inhibitor is used in this system, thus providing other data other than a 1.5x upregulation will provide strength to the data.

4. Please provide at least a table for DEGs, the figure 2c do not provide functional information to readers.

Author Response

I am extremely grateful for your hard work and critical suggestions. In this reply, I am sending my revised manuscript, which has been changed as suggested.

Reviewer 2 Report

Chen et al. have organized well written manuscript. Here, there are pieces of concerns that could be improved.

Major points:

1. Issues with figure presentation:

Fig. 4 and 7: Authors have showed representative photos. However, the ratio of 5hmC/5mC should be discussed with the data that exhibit the fluorescent signals of 5hmC and 5mC, simultaneously. This is important to understand the flux of cytosine modification in the cell.

Probably, authors use rabbit antibodies of both 5hmC and 5mC, which hindered the simultaneous measurement of 5hmC/5mC. Is there any reason for this strategy?

2. Issues with figure presentation:

Fig. 4: The signals of 5hmC and 5mC can be seen peripherally in the cytoplasm of oocyte. Sometimes, the signal seemingly condensed (maybe also not appropriate for quantification). Are there technical difficulties to keep oocytes at GV? Or, dose the GV of pig oocytes normally localize at the peripheral in the cytoplasm?

Minor points:

1. Issues with figure presentation

Fig. 2A: The concentration of Bobcat339 would be mistyped.

2. Specific comments

Line 230: Although authors explained DEG in method section, authors would better first spell out DEG here.

Line 375: The number of embryos used should be described in 4.9.

3. Typo

Line 212: The “oocytes” would be “groups”.

Line 323 and 326: “insulin-transferrin-sodium” may be “insulin-transferrin-selenium”.

Author Response

(The authors gave the same response as above.)

Round 2

Reviewer 1 Report

Congrats to authors for their work.